# Factors influencing the use of therapeutic footwear in persons with diabetes mellitus and loss of protective sensation: A focus group study

Athra Malki[1]*, Gijsbertus J. Verkerke[1,2], Rienk Dekker[1], Juha M. Hijmans[1]

1 Department of Rehabilitation Medicine, University Medical Center Groningen, University of Groningen, Groningen, Groningen, The Netherlands, 2 Department of Biomechanical Engineering, University of Twente, Enschede, Overijssel, The Netherlands

* a.malki@umcg.nl

## Abstract

### Background

Persons with diabetes mellitus (DM) and loss of protective sensation (LOPS) due to peripheral neuropathy do not use their therapeutic footwear (TF) consistently. TF is essential to prevent foot ulceration. In order to improve compliance in using TF, influencing factors need to be identified and analyzed. Persons with a history of foot ulceration may find different factors important compared with persons without ulceration or persons who have never used TF. Therefore, the objective of this study was to determine factors perceived as important for the use of TF by different groups of persons with DM and LOPS.

### Method

A qualitative study was performed using focus group discussions. Subjects (n = 24) were divided into 3 focus groups based on disease severity: ulcer history (HoU) versus no ulcer history (no-HoU) and experience with TF (TF) versus no experience (no-TF). For each group of 8 subjects (TF&HoU; TF&no-HoU; no-TF&no-HoU), an online focus group discussion was organized to identify the most important influencing factors. Transcribed data were coded with Atlas.ti. The analysis was performed following the framework approach.

### Results

The factors comfort and fit and stability/balance were ranked in the top 3 of all groups. Usability was ranked in the top 3 of group-TF&noHoU and group-noTF&noHoU. Two other factors, reducing pain and preventing ulceration were ranked in the top 3 of group-TF&no-HoU and group-TF&HoU, respectively.

### Conclusion

Experience with TF and a HoU influence which factors are perceived as important for TF use. Knowledge of these factors during the development and prescription process of TF

**Data Availability Statement:** The authors would like to make all anonymized data of the focus group publicaly available after acceptance of this article.

The data will be available on DataverseNL and will be assigned with a digital object identifier (DOI).

**Funding:** JMH received the funding for this work by EIT Health: https://eithealth.eu/. The Grant number was 211018. This work was part of the research project titled: 'IndiRock'nSole'. The funder had no role in the study design, data-collection and analysis, decision to publish, or preparation of the manuscript.

**Competing interests:** The authors have declared that no competing interests exist.

may lead to increased compliance. Although the main medical reason for TF prescription is ulcer prevention, only 1 group gave this factor a high ranking. Therefore, next to focusing on influencing factors, person-centered education on the importance of using TF to prevent ulcers is also required.

## Introduction

Diabetes Mellitus (DM) is one of the most common chronic conditions worldwide, affecting 422 million people [1]. The lifetime risk of developing a diabetic foot ulcer (DFU) in this population can be as high as 25%. Approximately 85% of diabetes-related lower extremity amputations are preceded by a DFU [2]. Therefore, DFUs are considered a major concern in healthcare, both from a quality of life and an economic perspective [3]. The largest risk factor for developing DFU is peripheral neuropathy [4, 5]. Sensory neuropathy causes insensitivity, leading to loss of protective sensation (LOPS). Additionally, a limited range of motion, atrophy of the small muscles in the foot, and changes in foot structure due to DM cause elevated plantar pressure. Repetitive stress on the areas with high plantar pressure leads to small wounds that may remain unnoticed due to the neuropathy, resulting in DFUs [6].

According to the International Working Group on the Diabetic Foot (IWGDF) guidelines on the prevention and management of the diabetic foot, persons with DM with a high risk of developing ulcers need to use therapeutic footwear (TF) in order to offload high plantar pressure [7]. The risk of developing DFUs can be reduced significantly by using offloading footwear for most of the day [8, 9]. However, several studies have shown low adherence to use of TF in persons with DM [10–12]. In order to improve adherence and prevent further deterioration of the condition of the foot, it is important to increase knowledge and awareness among persons with DM as well as to identify factors that are perceived as important for the use of TF [11]. This information can be used for the (future) development of TF and further improvement of the prescription process.

In a review published in 2016, only 6 quantitative observational studies were found to investigate adherence to use of TF in persons with DM [13]. Most studies in the review reported associations of different factors with adherence: foot deformity and minor amputation, perceived severity of the foot condition, TF appearance, body mass index, diabetes type, and age [13]. Due to the limited number of studies and inconsistent outcomes, it remains uncertain which factors influence adherence to use of TF. The inconsistent outcomes might be explained by the limited attention paid by researchers to the distinction between group and individual-level predictors of adherence. The heterogeneity of the group of persons with DM and foot complications likely explains the strong variation between individuals in TF use, thereby leading to predictors of adherence to only exist on individual or subgroup level [13].

A qualitative study is a suitable means to generate new knowledge on factors that are perceived as important for the use of TF (in different groups of persons with DM and LOPS). This type of methodology is not limited by pre-specified questions and can therefore identify the most important factors as well as the underlying perceptions, experiences, and wishes of participants regarding use of TF [14]. Focus group discussions with room for interaction will likely result in a fruitful discussion where ideas emerge that provide in-depth insight into the topic [15].

Since the population with DM is heterogeneous, in this study different groups of persons with DM and LOPS are distinguished, based on severity of the disease (HoU versus no-HoU)

and experience with TF. It is expected that persons who have experienced the impact of an ulcer (long healing time and necessity of good wound care) [16, 17]) will have different motivations for using TF compared with persons without ulcers. Other influencing factors are duration of TF use, positive or negative experiences with TF use, and type of footwear used. The aim of this study is to provide insight into the factors perceived as important (weight) by different groups of persons with DM and LOPS when using TF (based on severity of the disease and experience in using TF).

## Method

The Medical Ethics Committee of the University Medical Center Groningen (UMCG) declared that the study falls outside the scope of the Dutch law on Medical Research involving Human Subjects (WMO). Therefore, no formal approval was required from the committee (METc 2020/524). The required legal acts and/or guidelines, the Medical Treatment Agreement (WGBO), General Data Protection Regulation (GDPR) and, codes of conduct of the FEDERA (Federation of Medical Scientific Institutions) were followed.

### Subjects

Persons who visited physicians working at the Department of Rehabilitation Medicine of the UMCG or outside the UMCG, such as general practitioners or pedorthists (e.g., OIM Haren, the Netherlands) were recruited for this study. Subjects were also recruited from previous research (METc 2018/240), social media (e.g., Facebook and LinkedIn) and digital platforms (e.g., Diabetes Vereniging Nederland, Diabetesfonds, and Diabetes Trefpunt Nederland). Subjects were included in this study if they were diagnosed with DM, diagnosed with LOPS (self-reported), aged 18 years or older, able to speak Dutch, able to answer general questions, and able to attend the digital focus group discussion. The exclusion criteria were: having pain as the main complaint or not being able to walk independently. A purposive sample of eligible subjects who met the criteria were either contacted by phone, email or both and received an information letter with a written informed consent form. Eligible subjects were divided into 3 groups (8 subjects per group) based on disease severity (HoU versus no-HoU) and experience in using TF (experience versus no experience). For each group, additional group-level-related selection criteria were added. These criteria are shown in Fig 1.

### Study design and data collection

A qualitative study was performed consisting of 3 focus group discussions. The Consolidated Criteria for reporting Qualitative Studies Checklist (COREQ) was used for reporting the outcomes (see S1 File for COREQ Checklist) [18].

Prior to the focus group discussions, a self-reporting questionnaire was sent to all (24) subjects to gain knowledge about their disease and TF use (number of days per week and hours per day). Information on LOPS, foot deformities, HoU, prescribed devices, and months of TF use was retrieved. Additionally, the questionnaire was used to determine characteristics such as gender, age, body weight, and height. Due to the COVID-19 pandemic, physical measurements were not possible, which is why the information on the LOPS (diagnosed by a physician in a neurological assessment prior to this study), foot deformities (diagnosed by a physician prior to this study), body weight, and height was self-reported. In addition, the focus group discussions could not be held in person due to the COVID-19 pandemic. Therefore, the group discussions were held online using Microsoft Teams©.

Prior to the online meetings, all subjects received a letter with instructions on how the discussion was to be organized. If they had questions or needed more guidance, the first author of

**Fig 1. Inclusion and exclusion criteria for the 3 focus groups.** *TF = (orthopedic shoes (OS), semi-orthopedic shoes (semi-OS), or adaptation to ready-made shoes).

this study (AM) could be contacted prior to the focus group discussion. When joining the online meeting, subjects were informed (again) that the discussions were audio and video recorded and that they had to reaffirm consent. Thereafter, the moderator explained the purpose of the study, followed by a brief introduction of the subjects and the discussion round.

An interview guide (see S2 & S3 Files) with open-ended questions was developed. The questions were related to factors/subthemes that have been reported to (likely) influence a person's decision to use TF [11, 19–21]. Also, some general questions based on experience with TF and other topics (see S2 & S3 Files) were discussed during the focus group discussions [22]. The questions (order and formulation) were analyzed and discussed several times by the researchers and moderator(s) of this study in online meetings.

The total duration of each focus group discussion was 2 hours (including a 10-minute break). After the final focus group discussion, all recordings were analyzed.

## Moderator(s) and data coders

The moderator was an independent researcher (LK, PhD) experienced in conducting qualitative research. The assistant moderator, a PhD student (first author, AM) at the Department of Rehabilitation Medicine, UMCG supported the main moderator together with another researcher (RG, Human Movement Sciences Master student, University of Groningen) in logistical matters. An experienced qualitative researcher (last author, JMH) was also present at the focus group discussions. His role during the discussions was to ensure the focus group discussions proceeded according to plan and to answer questions from subjects. All of the researchers present during the focus group discussion were not acquainted with the subjects in advance.

## Data analysis

The recordings of all 3 discussions were transcribed verbatim by the first author. All subjects were given a coding number to preserve anonymity. Analysis of the transcriptions was performed with the Atlas.ti software package (Scientific Software Development GmbH, version 8.4.5 for Windows or Mac). The analysis process described below consists of different phases following the framework approach [23].

1. Read and re-read 3 transcriptions of the 3 focus group discussions.

2. Code 10% of 1 transcription (AM) and evaluate the initial codes (JMH).

3. Consensus meeting between (AM) and (JMH) to discuss initial codes and identify themes, subthemes, and factors (= code belonging to a certain subtheme).

4. Revision of the initial codes of 1 transcription and form a preliminary version of the thematic framework (AM).

5. Consensus meeting between (AM) and (JMH) to discuss the revised codes and the thematic framework.

6. Open coding first 10% of the 2 other transcriptions (AM) and evaluate the codes (JMH) and the thematic framework.

7. Consensus meeting between (AM) and (JMH) to discuss the new and revised codes and the thematic framework.

8. Code remaining 90% of 1 transcription (AM).

9. Consensus meeting between (AM) and (JMH) to discuss new and revised codes and the thematic framework.

10. Code remaining 90% of the 2 other transcriptions (AM). During this phase, data saturation was confirmed since no new themes, subthemes, and/or factors had to be added.

11. Consensus meeting between (AM) and (JMH) to discuss the final thematic framework.

12. Summarize quotes on the subthemes/ and or factors that were extensively discussed (AM) (Atlas.ti showed the frequency of the mentioned subthemes and factors) during the focus group discussions.

13. Consensus meeting between (AM) and (JMH) to discuss summaries on subthemes and/or factors.

The focus group discussions were held in Dutch. The data analysis (transcription and coding) was also performed in Dutch. All quotes presented in the result section were translated from Dutch to English (by a Dutch-speaking person).

## Results

A total of 24 subjects with DM and LOPS participated in 3 focus group discussions, with each group consisting of 8 subjects. The mean ages were 64, 63.9, and 63 years in group-noTF&no-HoU, group-TF&noHoU, and group-TF&HoU, respectively. The male-to-female ratio differed for each focus group. Characteristics of the subjects are shown in Table 1 (see S1 Table for the extended non-diabetes related comorbidities per focus group). This information was gathered from the questionnaire sent out to the (24) subjects. Other information gathered from the questionnaire was related to subjects' knowledge of their condition (cause of DFU) and whether they perceive TF as a solution for their complaints. Additional information on knowledge about reasons for TF prescription and types of footwear that exist became apparent during the start of the focus group discussion. This information can be found in Table 2.

### Results of the ranking process and most frequently coded factors/subthemes

During the focus group discussions, subjects were asked to rank the influencing factors from most (= highest rank) to least (= lowest rank) important. Data analysis followed after the focus

**Table 1. Characteristics of the subjects of the focus group discussions.**

| Characteristics | Group-noTF&noHoU n = 8 | Group-TF&noHoU n = 8 | Group-TF&HoU n = 8 |
|---|---|---|---|
| Age (years ± range) | 64 ± 25 | 63.9 ± 26 | 63 ± 17 |
| Sex (male/female) | 5/3 | 3/5 | 6/2 |
| Comorbidities (non-diabetes related) | | | |
| Hypertension | 4 | 4 | 4 |
| Arthrosis (back/hip/knee/wrists) | 2 | 5 | 1 |
| Heart failure/arrhythmia | 0 | 3 | 3 |
| Sleep apnea | 1 | 2 | 1 |
| Other | 3 | 6 | 3 |
| Type of current TF (number of pairs of TF) | | | |
| No TF | 3 | 0 | 0 |
| Inlays | 5 | 1* | 0 |
| Adaptation to ready-made shoes | 0 | 1 | 0 |
| Semi-OS | 0 | 2 | 2 |
| OS | 0 | 4 | 6 |
| Time since prescription of current TF (years ± SD) | 15.0 ± 9.5 | 3.3 ± 1.7 | 3.9 ± 3.7 |
| Use of current TF | | | |
| Days per week (number of days ± SD) | 5.8 ± 2.5 | 6.0 ± 2.6 | 7.0 ± 0.0 |
| Hours per day (number of hours ± SD) | 1.7 ± 1.2 | 7.1 ± 4.4 | 9.4 ± 3.3 |
| Walking ability (distance) | | | |
| 0 m | 0 | 0 | 0 |
| 0–10 m | 0 | 0 | 0 |
| 10–50 m | 1 | 0 | 0 |
| 50–200 m | 0 | 1 | 1 |
| 200 m– 1 km | 0 | 4 | 4 |
| More than 1 km | 7 | 3 | 3 |

Group-noTF&noHoU: group with no therapeutic footwear and no history of ulceration.

Group-TF&noHoU: group with therapeutic footwear and no history of ulceration.

Group-TF&HoU: group with therapeutic footwear and history of ulceration.

TF: therapeutic footwear, HoU: history of ulceration, semi-OS: semi-orthopedic shoes

OS: orthopedic shoes.

*Used semi-OS and OS in the past; however, they felt too heavy, which is why the subject returned to using inlays (this was also the case when the focus group discussion took place).

group discussions. During the data analysis, additional factors (and subthemes and themes) were coded (see S4 File for the final framework). These codes do not have a ranking because they were coded after the focus group discussions took place. The input of subjects on factors that were coded often and/or ranked during the focus group discussion was summarized per group. In Table 2, results of the data analysis are shown. It is important to note that certain definitions (i.e., usability) were further divided into different factors during the data analysis process. For example, the factors appearance and prescription process are also ranked in Table 2. Both are subthemes, but in order to make the discussion easy to follow, these subthemes are referred to as factors as well. All other factors shown in Table 2 can be grouped under the subthemes use, effectiveness, and usability of TF. The input given by each group is shown in Table 2 and further outlined in the section below. It is important to note that input of group-noTF&noHoU was based on future use of TF and that this group had no prior experience with TF (except for inlays).

**Table 2. Knowledge on the reasoning behind and function of TF and the list of factors that are perceived as important by the different groups of subjects with DM.**

| Knowledge on the reasoning behind and function of TF | | |
|---|---|---|
| Subjects in all groups seemed to have sufficient knowledge about the complications of having diabetes. Group-noTF&noHoU and Group-TF&HoU mentioned neuropathy and pressure spots as complications. Callus formation, pain (Group-noTF&noHoU), foot deformities (Group-TF&HoU), and ulcers (all groups) were also mentioned as diabetes-related complications. All groups seemed to know TF is prescribed to prevent DFUs. Other benefits of using TF were also mentioned, such as walking with less effort and pain, better stability, maintaining foot health (Group-noTF&noHoU), reducing pain, and improving balance (Group-TF&noHoU). | | |

| Factors perceived as important | | |
|---|---|---|
| Rank[a] | Group-noTF&noHoU | Group-TF&noHoU | Group-TF&HoU |
| 1 | **Comfort and fit** | **Comfort and fit** | **Comfort and fit** |
| 2 | **Stability/balance** | **Reducing pain** | **Stability/balance** |
| 3 | **Usability (weather/material/activity)** | **Stability/balance**[*] | **Preventing ulceration(reducing pressure)** |
| 4 | Appearance | **Usability(weather/material/activity)**[*] | Reducing pain |
| 5 | Donning and doffing | Weight | Walking quality (improving walking distance) |
| 6 | Weight | Walking quality (improving walking distance) | Usability (weather/material/activity) |
| 7 | Preventing ulceration | Appearance | Donning and doffing |
| 8 | Reducing pain[*] | Donning and doffing[*] | Prescription process by physician/pedorthist |
| 9 | Walking quality[*] (improving walking distance) | Prescription process by[*] physician/pedorthist | Appearance |
| 10 | Prescription process by[*] physician/pedorthist | Preventing ulceration[*] | Weight |

| Additional factors coded during data analysis (random order, not ranked) | | |
|---|---|---|
| Durability | | |
| Location (indoors/outdoors, special occasion, work etc.) | | |
| Type/number of pairs | | |
| Sole thickness | | |

Group-noTF&noHoU: group with no therapeutic footwear and no history of ulceration.

Group-TF&noHoU: group with therapeutic footwear and no history of ulceration.

Group-TF&HoU: group with therapeutic footwear and history of ulceration.

TF: therapeutic footwear, HoU: history of ulceration.

[a]The factors set in bold are the factors the subjects in the 3 different groups ranked in the top 3. Ranks 4 to 10 are based on the number of times the factors were mentioned as important during the focus group sessions.

[*]Same ranking of different factors per group.

## Main findings

**Group-noTF&noHoU factors belonging to the subthemes use, effectiveness, usability, appearance, and prescription process of (non-) TF.** *Comfort and fit*. During the discussion, subjects mentioned the importance of having comfortable and well-fitting footwear, not only for walking long distances but also for indoor use. Subjects in this group ranked comfort and fit of TF as the most important factor influencing their (future) use of TF. According to this group, comfortable footwear should meet the following criteria: soft and cushioned, light, and seamless.

> D1.6: . . .but what is important, uh that is uh that it has decent shock absorption, that it is a bit soft, uh so that it has enough cushioning. I think that is very important.

*Stability*. Stability when using TF was not discussed extensively; however, it was ranked as the second most important factor. According to one subject (D1.7), it is not possible to make a move without using sturdy footwear because this provides the required support and stability during walking. The same subject used shoes the whole day for fear of falling.

*D1.7: . . .. If I take them off, then I cannot take a single step forward. Even even with with the balance and so on, eh then it is done. I already fell a couple of times. Because I just, well, maybe not necessarily have stability issues, but I am not able to stand up straight.*

*Weather and material.* The weather and material of TF were the third most important factors. Again, these factors were not discussed extensively. Only one subject (D1.2) mentioned that TF needs to have cushioning and has to be made of breathable and waterproof material. Another subject (D1.5) mentioned the added value of having TF in the form of sandals for the summer.

*D1.5: But what I am also very curious about is when you have closed-toe shoes for the winter, what about summer shoes. Are there sandals that can be adapted to uh to an OS. I am actually very curious about that. For example, with an open uh toe or uh, well that seems dangerous for a diabetic by the way. And I was not allowed to wear sandals for a long time, by the way, but secretly I do wear sandals in the summer when it is very hot, as it can get really hot nowadays in the Netherlands. I put my inlays, my my orthotics in them.*

*Appearance and sole thickness.* Appearance was ranked just outside the top 3 of most important factors. However, in contrast to some of the aforementioned and higher ranked factors, this particular factor was discussed extensively during the focus group discussion. The factor sole thickness was not ranked during the focus group discussion; however, it was coded during the data analysis and was linked to appearance. Most subjects mentioned that although appearance of TF is important for their future decision to use TF, it is less important than using comfortable footwear. However, when the researchers' prototype shoes (sporty look with thick midsoles, see S1 Fig for prototype shoes) were shown, most subjects did not like the shoes and mentioned they would not use them unless they had to. They felt the shoes were not appropriate for more formal occasions (e.g., visits, weddings). According to some subjects, the prototype shoes only met the standards for outdoor use, such as walking. Only one subject (D1.7) was willing to use the prototype shoes; however, even this subject preferred to change the colors of the shoes.

*D1.8: Uh so as to the appearance of the shoes. I think uh that with outdoor shoes, they should look a bit decent. Because you have to go to occasions where you have to look a little bit more decent sometimes, so to speak. But uh with shoes that I only use indoors, then it does not really matter to me, as long as they are comfortable. It does not matter what they look like.*

*Donning and doffing and weight.* The factors donning and doffing and weight of the shoes were not rated highly. Both factors were also not discussed extensively within this group. Only 2 subjects (D1.2 & D1.5) mentioned that the donning and doffing process should be easy to perform. Three subjects (D1.2, D1.5 & D1.8) mentioned the importance of lightweight footwear, especially for indoor use (D1.5).

*D1.5: And uh when you are indoors and yes I am also at an age where I do not have a very busy life anymore. So I go outside for an hour or so every day, but other than that, I am busy indoors. And if you always have to wear those heavy, somewhat stiff shoes, well, I would not like that.*

*Preventing ulceration.* Preventing ulceration was not ranked highly. Nevertheless, it was discussed extensively during the focus group discussion. Subjects were aware of the important

role of TF in preventing ulceration. They also mentioned that feet stay healthy longer when using TF, wide feet in particular (D1.7). One subject (D1.5) also mentioned that TF should be seamless in order to prevent shear stress, which may result in wounds.

> *D1.7: The width of the foot, yes I always have uh well problems with that. It it pinches too much. And then those pressure points emerge uh uh emerge on my foot.*

*Reducing pain, walking quality, and prescription process.* The factors reducing pain, walking quality, and prescription process were considered the least important factors. The factors pain reduction and walking quality were also not discussed extensively. In general, subjects mentioned that using TF could lead to walking with less effort and pain. Contrary to the 2 other factors, the factor prescription process was discussed extensively. Subjects mentioned they want to be involved during the prescription process and that physicians and/or pedorthists should listen to their wishes and give them feedback on the existing possibilities. They wanted to be able to express their opinion on the appearance of TF and discuss the level of comfort and stability of the footwear. Finally, subjects also found it important for their prescribing professional to be knowledgeable and experienced.

> *D1.4: Well I uh I would indeed first find out who can help me and uh also consider the the certifications that someone has. Uh I would very much like to contribute to the whole process. Also uh that someone looks at your needs and wishes, what are the requirements of a shoe. Not only in terms of appearance but also in terms of comfort. I would find that very pleasant.*

*Location, type/number of pairs, and durability.* The location (e.g., indoors/outdoors, special occasions, work), type/number of pairs, and durability were not ranked during the focus group discussion; however, these factors were coded during the data analysis. The factors type/number of pairs and durability were not discussed extensively in this group. One subject (D1.2) wished to have water resistant TF. Another wanted TF that can be used during the summer period and is also suitable for use indoors (D1.5). With regard to location, subjects used inlays/sturdy footwear when going for a walk, travelling to work, or staying at home. For indoors, some subjects put their inlays in their slippers. Others mentioned they had to use sturdy footwear. Some bought a pair for indoor use only. Two subjects (D1.2 & D1.5) pleaded for the development of indoor TF.

> *D1.8: No because I have special shoes for both indoors and outdoors and I use them uh all day long. When I'm outside, I really use my outdoor shoes, which I can walk on just fine. But when I am indoors, I also use shoes because slippers and bare feet just do not work at all.*

**Group-TF&noHoU factors belonging to the subthemes: Use, effectiveness, usability, appearance, and prescription process of TF.** *Comfort and fit.* The outcomes related to comfort and fit were similar to the group-noTF&noHoU. In the group-TF&noHoU, one subject (D2.5) mentioned that TF improves posture (resulting in less back pain) and makes it possible to walk long distances pain free. Some subjects (D2.6 & D2.8) emphasized the importance of feeling the surface underneath their feet, which is why they wanted footwear with thin midsoles. However, not every pebble should be felt, which is why (D2.8) mentioned this issue could be resolved by TF with inside cushioning.

> *D2.8:..Yes. I uh when I was still wearing non-therapeutic shoes then uh I could hardly walk due to the pain because I felt every pebble on the way. And then the pedorthists said, he/she*

said: 'We can take care of that by elevating the shoe from the inside and place an inlay in the shoe'.

*Reducing pain*. Pain reduction was ranked as the second most important factor. Again, similar to the group-noTF&noHoU, subjects stated they feel less pain when using TF. Some subjects felt pain reduction was caused by the added ankle stability. Other reasons for pain reduction were having fewer corns and a better posture. One subject (D2.8) was able to walk longer distances due to pain reduction.

*D2.8: I can walk further with uh my OS on of course, but as I just said. I have 2 new knees and a new hip so that does limit me in my walking; however, that is with no pain in my feet, for sure.*

*Stability, weather, and material*. The factor stability, specifically related to the weather and material of TF, was ranked in third place. Subjects experienced more ankle stability due to the use of TF. However, one subject (D2.3) mentioned that this is only the case for walking on flat surfaces, which is why a solution is needed for walking on unequal surfaces. In this group, TF was used more outdoors than indoors since the stability TF provides is necessary for outdoor use. Indoors, subjects kept their balance by walking barefoot or by using slippers that offer enough stability. The outcome for weather and material was similar to group-noTF&noHoU.

*D2.3: Yes. uhm uh on uneven surfaces I walk off balance. Uhm I notice that I often have to adjust, also often sprain my ankle . . .uhm so just on the street so to say then I really walk in balance. That is very nice, but on uh on really uneven surfaces it does not work well.*

*Weight*. The factor weight was ranked just outside the top 3 of most important factors. The group found it important to have lightweight TF for easy walking, particularly in case of weak muscles according to one subject (D2.6). One subject (D2.3) did not use TF indoors or during ball sports because of its heavy weight. This would be different if the heavy footwear could be replaced with lightweight footwear.

*D2.6: Uh I used uh semi-OS years ago. Those became too heavy to use at one point, so I stopped using them. We tried uh real OS and those were fully custom made. But when I put them on, I stood and I kept standing and I absolutely could not walk on them, so uh they are in the attic and 2 years ago we tried to have OS custom made again and were unsuccessful.*

*Walking quality, appearance, and sole thickness*. Although these factors did not make the top 3, subjects did give some input. Similar to the group-noTF&noHoU, sole thickness was discussed in relation to appearance of TF. Outcomes for walking quality were similar to group-noTF&noHoU. Some subjects (D2.5 & D2.8) mentioned better posture and pain reduction, which improves walking quality. Consequently, they were able to walk longer distances with their TF. Outcomes for appearance were also similar to the group-noTF&noHoU.

*D2.5: Well if I uh my my uh my boots are on so to say, not my (therapeutic) shoes, then I uh walk for about 15 minutes and after that uh then I start to lose my balance. Then it becomes uh and if I do use my (therapeutic) shoes, then uh I walk for half an hour to an hour without uh losing my balance.*

*Donning and doffing, prescription process, and preventing ulceration*. These 3 factors were ranked as the least important factors to influence the decision to use TF. They were also not

discussed extensively. Subjects mentioned that although they find these factors important, they are not as important as other, previously mentioned factors. Subjects mentioned the donning and doffing process should be easy to perform. Some subjects (D2.7 & D2.8) suggested TF with a zipper or velcro instead of laces. One subject (D2.5) mentioned that laces are acceptable as well and that you have to get used to laces. In general, subjects were positive about the time and effort the donning and doffing of TF takes. Regarding the prescription process, some subjects (similar to group-noTF&noHoU) mentioned the importance of being involved in the process and feeling seen and heard by the physicians and/or pedorthists. They also stressed the importance of well-fitting TF. One subject (D2.6) did not use TF because it was ill-fitting. The outcome of preventing ulceration was similar to group-noTF&noHoU.

*D2.7: Yes I I have uh uh for years, so first I had uh as far as that German manufacturer goes it was 'Finn Comfort' uh and uhm I always get uh from them and also this time just Velcro and it can be also pulled up tight, just as tight as you actually want. So uh yes I like that very much. My stomach is still a little bit in the way, so tying (shoelaces) is a uh a disaster. \*laughs\**

*D2.5: Yes what what I actually think is very important is that they fit well but also that they are seamless.. . . So that they are finished seamlessly and the pressure is equally distributed. That has turned out to be uh, very important.*

*Location, type/number of pairs, and durability.* As mentioned before, location, type/ number of pairs, and durability were coded during the data analysis. During the discussion, subjects mentioned they use their TF outdoors (e.g., for work, car rides). However, some (D2.5 & D2.8) did not use their TF in the forest or at the beach. They want to keep their footwear clean and intact because they are only prescribed a limited number of pairs within a certain time frame. This makes them very cautious, and a number of subjects expressed dissatisfaction with the prescription limitation. They also noticed a lack of TF suited to different weather situations (therapeutic sandals do not yet exist). In addition, some subjects also did not use their TF indoors because they find slippers more comfortable (especially after using TF outdoors for the majority of the day).

*D2.8:. . .Only indoors, in the house, yes then I wear slippers and of course that is not that is not so good. Since you do not have as much support but uh well so far it is going well. And in the summer I like to use uh the Birkenstock slippers. They have a reasonably decent foot bed, but of course I have no support uh for my ankle. But if I develop issues, I put them (TF) back on, similar to what the other subject also just said about walking outdoors. We like to go to Texel on vacation and walk on the beach there, and during these moments I feel it is a pity to use the (OS) shoes. So I either use my slippers when the weather is good or or I put on old shoes and think: well, this will have to do for now. Because it is a pity that the (OS) shoes get so ugly, especially from the seawater.*

**Group-TF&HoU factors belonging to the subthemes: Use, effectiveness, usability, appearance and prescription process of TF.** *Comfort and fit.* Outcomes related to comfort and fit were similar to group-noTF&noHoU. The subjects in this group (group-TF&HoU) also stressed the importance of comfortable footwear in relation to improving balance and experiencing no/less pain sensations.

*D3.7: But I do notice that walking comfort is a lot a lot better compared with with regular shoes. Less chance of tripping and stumbling.*

*Stability*. The factor stability was ranked in second place. A number of subjects mentioned they often fell prior to using TF and that this changed after they started using TF. TF provides ankle stability (also mentioned in the 2 other groups), which is why users walk with more confidence, have fewer problems walking on different surfaces, have fewer back problems, and are able to use the stairs without fear of falling. Some subjects mentioned they use TF indoors as well for these reasons.

*D3.6: I uh I think I told you that that I did uh fall a few times. But not any longer because of those shoes I have been wearing since March for almost 16 hours a day and that uh works perfectly.*

*Preventing ulceration*. Preventing ulceration was ranked in third place. During the discussion, subjects mentioned the reason for prescribing TF is to prevent ulcers. All subjects mentioned they have not developed any new wounds since using well-fitting TF.

*D3.1: Uh first I had low OS uh that fit well, also due to pressure spots and such, but about 5 years ago I began to have strange falls due to loss of sensation in my feet and whenever I stepped over a twig or a pebble I would strain my ankle. Subsequently, I received high ankle supportive uh OS. Afterwards it did not happen again.*

*Reducing pain*. This factor was ranked just outside the top 3. The outcomes for this factor were similar to the other groups regarding reduced pain sensation (group-noTF&noHoU & group-TF&noHoU) and walking long distances (group-TF&noHoU).

*D3.7: In my case, I do notice that I am a bit more pain free and therefore walk a bit more comfortably. Uh because besides those pressure spots, I also have rigid uh rigid toes that do not move if I load them with the wrong uh shoe. Then yes it becomes painful and because it is less painful with TF, I walk a bit easier and a bit more.*

*Walking quality*. Although walking quality was ranked outside the top 3 and was also not discussed extensively, it was still ranked relatively high by subjects in this group. Similar to group-TF&noHoU, one subject (D3.5) could walk longer distances when using TF. This subject also mentioned that walking quality improved due to better posture, which in turn reduced back problems.

*D3.5: this shoes uh the winter shoes I currently have are also higher and a bit chunky but they work really well and uh I can walk rounds of 5–7 kilometers or sometimes 10 kilometers with the dog. So I think they are fine and comfortable and I just uh I just can carry on. And well, I walk straighter so it is also better for my back.*

*Weather, material, location, type/number of pairs, and durability*. Despite the low ranking of the factors weather and material, the subjects did have some discussion points related to these factors. The 3 other factors, location, type/number of pairs, and durability, were often coded during the data analysis and were linked to weather and material during the discussion in this particular group. In general, subjects wished for water resistant TF that they could use during rainy weather or when going to the beach or swimming pool. They also wanted to have footwear made of thinner materials for the summer. Although all-weather TF meets these criteria, subjects (similar to group-TF&noHoU) mentioned the prescription limitation. All-weather footwear has a sporty look and is not appropriate for every occasion; therefore, subjects often

opt for other types of TF. As a result, they do not use their TF as often as they would like to. To protect the TF, one subject (D3.7) does not use TF while dog walking. However, some subjects cared less about protecting their footwear and continue to use it during walks. The same difference in opinion was found between subjects about wearing TF to work or during special occasions. Some use TF during working hours, whereas others said that it depends on their function and whether formal footwear is required. Some subjects do not mind wearing TF during special occasions; however, one subject (D3.5) in particular mentioned having difficulties with wearing TF during visits. Regarding indoor use, most subjects do not use their TF indoors. They often use slippers with inlays and mentioned they wanted (similar to group-noTF&noHoU) TF specifically made for indoor use. Only one subject (D3.1) uses TF indoors due to several fall accidents. Another subject (D3.5) uses slippers instead of TF when going to the swimming pool. However, this subject also had some fall accidents related to not using TF.

*D3.7: The disadvantage is just, but that is my own choice, that I uh I have formal shoes for in the summer and now more sturdy shoes. But like someone else, who I just heard, I walk the dog a lot, and then I still use regular shoes uh that are weatherproof. . . It would have been better if I had chosen all weather shoes or something close to that for now. So yeah that is going to be a matter of waiting to to fix that. Or that the weather will be nice and I can wear the normal shoes. So something to think about is that how you can use the shoes is highly dependent on what what type of shoes you choose.*

*Donning and doffing.* The factor donning and doffing was also ranked low. Again, some discussion points were mentioned. Subjects observed that donning of TF takes time. However, they received tips and tricks from pedorthists to speed up this process and some of these were perceived as useful. One subject (D3.5) mentioned that during visits or when going for a swim, the donning and doffing remains a hassle. For this reason, subject D3.5 always uses slippers to the swimming pool, which have resulted in a few fall accidents.

*D3.5: so it is uh but indeed uhm just what you are saying about sports or swimming. In the summer, I swim in the outdoor pool a lot, and well, you use your slippers then because uh taking the socks and shoes off is a bit burdensome. So you also have a greater chance of falling. That also happened this year because I walked on slippers or other uh older shoes.*

*Prescription process.* Prescription process received a low ranking but was discussed extensively. Experiences with the prescription process of TF differed for each subject. Some were positive and said the fitting of their TF was spot on the first time; others were negative and mentioned a process of trial and error with an unsatisfying result. Some subjects observed a lack of consensus between the different pedorthists, and they wished they could have a second opinion. The result of the fitting process greatly depends on the experience and expertise of the pedorthists. One subject (D3.8) mentioned that during the fitting 2 pedorthists were involved, and that the result was satisfactory. Subjects in this group (similar to the other groups) also mentioned they want to be included in the prescription process of TF, and they wish for physicians and/or pedorthists to listen their problems in order to find a joint solution.

*D3.8:. . . you should also give some clear examples, sketch situations (for instance) in which you need the shoe..and I must say that they cater to that well. I now have uh 2 advisors uh who [work] together, and I must say that I do like it.*

*Appearance and sole thickness*. Appearance received a low ranking. A number of discussion points related to this factor were mentioned. Similar to the other groups, sole thickness was linked to appearance of TF. Subjects stated they find appearance important. In general, they reported being content with the appearance of their TF, but they also stressed that this is not the main reason for using TF. However, when they were shown the researchers' prototype shoes, which have a sporty look with thick midsoles (see S1 Fig for prototype shoes), approximately half of the group was not willing to use the shoe. They pointed out that the shoe with the thick midsoles look too chunky. In addition, one subject (D3.5) mentioned being afraid of toppling over because of the thick midsoles. The donning and doffing of the shoes with the thick midsoles was also perceived as inconvenient. Only 2 (D3.1 & D3.7) subjects were willing to use the prototype shoes if these could solve their foot problems, and one other subject (D3.3) was willing to use the shoes if they were comfortable.

> *D3.4: Uh well, I I would not use them, I mean, uh it might might have something to do with vanity, but I would just go for shoes with a normal thin soles that looks the most like non-therapeutic shoes. Uh these shoes, as the previous speaker indeed already mentioned, are from the past, shoes with a thick platform sole, you really would not see me walking on those.*

*Weight*. Weight of TF received the lowest ranking and was not discussed extensively. Subjects did have some discussion points. They reported their TF is not or only slightly heavier than non-TF, but the difference is not significant. According to some subjects, you get used to the weight of TF when using it more often and heavier TF provides more stability.

> *D3.3: So I uh do not feel the shoes are heavier now. The ones I that I actually got 10 years ago those were really heavy shoes but over the years they have gotten much lighter.*

## Discussion

This study aimed to provide insight into the factors perceived as important for the use of TF by different groups of persons with DM and LOPS. The 3 focus group discussions provided mixed results. First, the factor comfort and fit was ranked as the most important factor to influence the decision to use TF in all groups. The other factors were all ranked differently by the 3 groups. The factor ulcer prevention, which is the main reason for TF prescription, was ranked in the top 3 by just one group. This was the group with a history of ulceration (group-TF&HoU). The results are further outlined in detail in the sections below. As mentioned before, it is important to note that group-noTF&noHoU is the only group that had no experience with TF (except for inlays) and that all their opinions are based on the future use of TF.

As mentioned above, the factor comfort and fit was given the highest ranking in each group. This finding is in line with the research of Arts et al., in which 33.3% of the persons with DM (n = 145) reported comfort to have the highest priority of all aspects of footwear usability [11]. It is noteworthy that the subjects in this study chose comfort and fit as the most important factor because they are not able to feel the footwear properly due to LOPS. According to Arts et al., this might be explained by the fact that subjects related comfort more to walking comfort/walking quality instead of fit of the footwear [11]. However, this reasoning is not entirely in line with the results of this study. In this study, a distinction was made between the factors walking comfort/walking quality and (comfort and) fit. Subjects were asked to rank the factors comfort and fit and walking quality separately. However, it could be that persons interpret comfort and fit as having a broader definition than only the fit of the TF, which again is in

line with the study of Arts et al. and van Netten et al. that comfort (and fit) can be interpreted as more than one definition [11, 20].

The factor ulcer prevention got a low ranking by group-noTF&noHoU (second-to-last rank) and group-TF&noHoU (lowest rank). Group-TF&HoU ranked this factor third. The positive association between severity of the foot condition (Group-TF&HoU persons had poor foot health) and the importance of using TF is in line with previous literature [24–26]. The reason why this factor was not ranked in first place could be related to subjects' underestimation of their actual foot health. The study of Macfarlane and Jensen confirms this. They found that most persons with DM believed their foot health was better compared with other persons with DM, despite the fact that 62% had a history of foot complications [27]. Although a HoU resulted in a higher ranking of ulcer prevention, it was still not the most important factor. Consequently, foot complications alone do not motivate persons enough to increase their use of TF and thereby prevent ulceration. A recent study of Keukenkamp et al. analyzed the role of motivational interviewing to improve adherence to TF use [28]. Their study showed a positive association between introducing motivational interviewing during the initial prescription process and adherence to TF use. Although the effect of 1 session lasted for a short period of time only, introducing regular sessions of motivational interviewing could help to make sure persons do not underestimate the severity of their foot condition, understand the main reason for using TF, and stay motivated to use their TF.

The factor stability was another factor ranked in the top 3 by all groups. Group-noTF&no-HoU and group-TF&HoU ranked this factor as the second most important factor and group-TF&noHoU ranked this factor in third place. Persons with DM and LOPS can experience postural instability due to lack of somatosensory feedback, which makes it important to use footwear that does not deteriorate their balance even more [29–31]. Subjects in group-TF&noHoU and group-TF&HoU mentioned they experienced fewer stability issues when using TF compared with non-TF. Subjects in group-TF&HoU with some fall accidents prior to using TF no longer fell after they began using TF. Experiencing the benefits of having reduced stability issues and no fall accidents, seems to be a motivating factor for using TF. Most subjects in group-noTF&noHoU did not experience any falls; however, a possible fear of falling due to current (or future) stability issues with non-TF can also be a motivating factor for using TF.

The factor pain reduction was ranked in the top 3 by group-TF&noHoU. Group-TF&HoU ranked this factor just outside the top 3, and group-noTF&noHoU gave this factor the lowest ranking (least important factor). The difference in ranking between group-noTF&noHoU and the other 2 groups could be related to the impact of pain experience. During the focus group discussions, subjects from group-TF&noHoU and group-TF&HoU mentioned that using TF could reduce pain. Group-noTF&noHoU did not seem to have issues related to pain, and it is therefore not surprising they attached lower significance to this factor compared with the other 2 groups.

Group-noTF&noHoU and group-TF&noHoU ranked the factors weather and material in their top 3 of most important factors. Group-TF&HoU ranked these factors lower compared to the 2 other groups. This lower rank might be related to poorer foot.health. Other factors, closely related to weather and material, such as durability, location, type/number of pairs of TF were discussed as well. Because most subjects were mainly active indoors, it is a worrisome finding that TF was not worn indoors because of a perceived lack of TF made for indoors or other reasons [10, 32, 33]. Several companies have made TF for indoor use, according to Keukenkamp et al. However, none of these companies followed a systematic approach where they integrated the professionals' and users' perspectives [33]. This is why Keukenkamp et al. focused on designing TF for indoor use with the same biomechanical efficacy as TF that is intended for outdoor use [33]. They also assessed the users' expectations and needs regarding

TF for indoor use in their study. Introducing this type of TF could improve adherence to TF use, particularly because subjects in this study mentioned they were willing to use TF indoors if it was made for them [30].

The factor appearance was not ranked in the top 3 of most important factors in all groups. The ranking order also differed for each group. Group-noTF&noHo gave the highest ranking. This finding is not surprising, other studies have shown the importance of appearance in using TF in persons with DM [10, 34]. It seems that with an increasing severity of foot problems, the factor appearance becomes less important. All subjects mentioned that effectiveness (e.g., ulcer prevention) of TF was more important than appearance of TF. However, appearance was ranked higher than ulcer prevention in group-noTF&noHoU and group-TF&noHoU. Group-TF&HoU was the only group that ranked the appearance of TF lower than ulcer prevention. This finding is worrisome because it indicates that the value of using TF is only acknowledged after experiencing an ulcer.

In a previous study, the factors weight and donning and doffing were said to have an important role in adherence to TF use, especially indoors [33]. Contrary to earlier studies, where the importance of having lightweight TF was stressed, group-TF&HoU did not give it much of a significance (lowest rank) [11, 27, 34]. The 2 other groups ranked the factor higher than group-TF&HoU. According to group-TF&HoU, heavier TF does not pose any difficulties because it provides more stability. Since group-TF&HoU has less hinder from heavy footwear, it is likely that they were less frequently reminded of the weight of the footwear, and therefore did not give this factor a high ranking. Apart from the weight of the TF, subjects in all 3 groups mentioned they wanted TF with an easy donning and doffing process. Group-noTF&noHoU gave the highest ranking to this factor compared with the other 2 groups. This could be explained by the fact that the other groups are more used to donning and doffing and do not attach as much significance to this factor as the group with no prior experience.

Group-noTF&noHoU and group-TF&noHoU ranked the factor prescription process lowest. In contrast, Group-TF&HoU ranked this factor higher. This higher ranking may be explained by the fact this group experienced more issues during the prescription process compared with both other groups. Subjects in all groups did mention the importance of active listening by the physicians and/or pedorthists during the prescription process. Subjects in group-noTF&noHoU and group-TF&HoU also emphasized the importance of getting help from an experienced physicians and/or pedorthists during this process.

## Strengths and limitations

An important strength of this study is its qualitative design, which gave 3 small groups of subjects the opportunity to discuss different subthemes/factors. This set-up led to in an in-depth discussion that ultimately resulted in an overview of factors perceived as important for the use of TF. A qualitative design highlights the reasoning (experiences, perceptions, and wishes) behind the importance attached to factors that influence the use of TF. A quantitative design could be the next step in analyzing whether a larger population agrees with the ranking of the different factors and also whether factors are missing. Tenny et al. advocate the use of different types of designs (qualitative and quantitative) because they provide complementary information [35]. Another strength of this study is that it included a diverse group of subjects with regards to use of TF and severity of the disease. A group of 8 subjects was found to be sufficient for reaching data saturation. However, it is possible that groups of subjects with DM and LOPS from other national or cultural backgrounds would propose other factors. This study included Dutch subjects. In other countries, the health care system, insurance system, costs, accessibility of health care, and cultural background (e.g., family role, work, religion) could be

different. Therefore, generalizability of the data is applicable to the Dutch population but may not be applicable to other cultures and nationalities.

This study has some limitations. The factors that were ranked in this study emerged from the discussion based on the questions related to factors/subthemes that have been reported to (likely) influence a person's decisions to use TF [11, 19–21]. To the researchers knowledge, no framework exists that was specifically built to study the factors influencing footwear use in persons with DM and LOPS. As a consequence, the researchers might have focused on a too narrow range of (relevant) factors influencing the TF use. However, during the 3 focus group discussions, subjects did get the opportunity to introduce new factors if they felt some were lacking. None of the subjects added new factors; however, thereby increasing the probability that all relevant factors were addressed in this study. Additionally, due to the COVID-19-pandemic a neurological assessment was not performed to diagnose LOPS. Although subjects were not specially diagnosed with LOPS for this study, they were neurologically assessed by a physician who diagnosed persons with LOPS prior to this study. It was not possible to verify whether the subjects who were recruited via digital platforms and social media had indeed been diagnosed with LOPS by a physician. However, there was no reason for the subjects to be dishonest about their diagnosis since they did not benefit from participating in the study. The same goes to physical measurements to asses foot deformities. Due to the COVID-19 pandemic, the focus group discussions were held online. This can have potential disadvantages. Some subjects may be uncomfortable using technology or experience technical difficulties. Researchers cannot observe full body language of subjects and potential distractions may go unnoticed. This study aimed to reduce these disadvantages to a minimum. The researchers made sure to include subjects who had no difficulties with technology as well as subjects who lacked computer experience. The latter were guided by the researchers (and by their family) on how to participate online. Subjects who experienced technical difficulties (e.g., joining the meeting, video or microphone issues) during the focus group discussion were assisted by one of the researchers by telephone in order to solve their issue. Although full body language was not observed, the researchers could still focus on the persons' facial cues to gain some non-verbal information. Last, since subjects joined from home, they could be distracted by their environment. To prevent unwanted distractions, the researchers asked the subjects to sit in a quiet area of their house.

### Future recommendations

This study is the first qualitative study to provide an overview of factors perceived as important (and of their importance weight) for TF use by different groups of persons with DM and LOPS. This overview can be used during the prescription and the development process of TF. It offers physicians, pedorthists, and other professionals a better understanding of the factors that need to be taken into consideration when prescribing TF. Future research could focus on possible other missing factors and the development of a conceptual framework for comparing possible interactions between relevant factors [36]. This study also showed that ulcer prevention did not have the highest priority in the different groups, despite the fact that the subjects were aware that this is the main reason for TF prescription. To address this discrepancy, new education methods need to be introduced. Keukenkamp et al. introduced motivational interviewing as a new method to increase adherence to TF use [28]. Their study showed the design was successful in increasing the adherence to TF use. However, this was solely the case for a short period of time. A possible solution might be to introduce regular motivational interviewing sessions instead of only once (with 2 45-minute sessions in 1 week). Other suggestions to increase the use of TF were given by Jarl et al [26]. Their suggestion was to eliminate the

temptation to use non-TF and provide a reminder to use TF [26]. This can be achieved by keeping non-TF out of sight and keeping TF in plain sight in order to send out a visual cue to use the TF and reduce the effort to retrieve non-TF from somewhere else [26]. This solution could work; however, if the reason for not using TF is related to protecting it from bad weather conditions, another solution is needed. Although all-weather TF exists, apparently these shoes do not meet all criteria for use. Either some changes need to be made to all-weather shoes, or subjects should be prescribed a new pair of TF more often. The latter option is likely to increase the costs, which is a disadvantage. Nevertheless, the cost of wound care is higher than a new pair of TF. One of the findings of this study was that most subjects did not use their TF indoors because indoor TF has not yet been developed. However, as discussed before, Keuken-kamp et al. have presented a design for indoor TF following a multidisciplinary systematic design approach. Introducing this type of footwear could possibly increase indoor use of TF [33].

## Conclusion

Different experiences with TF as well as a HoU influence which factors are perceived as important for TF use. The factors comfort and fit and stability/balance were ranked in the top 3 of all groups. Taking these factors into account during the development and prescription process of TF may lead to increased adherence. The most interesting finding was that subjects only realized the importance of giving priority to ulcer prevention after experiencing an ulcer. Nevertheless, even with a HoU, subjects still did not rank this factor highly. This finding emphasizes that besides focusing on the factors that are perceived as important for TF use, person-centered education on the importance of using TF to prevent ulcers is also essential.

## Supporting information

**S1 File. COREQ Checklist.**
(XLSX)

**S2 File. Interview guide for the focus group session of group-noTF&noHoU.**
(PDF)

**S3 File. Interview guide for the focus group sessions of group-TF&noHoU & group-TF&HoU.**
(PDF)

**S4 File. Final framework of the focus group discussions.**
(PDF)

**S1 Fig. First version of the prototype shoes with varying thickness of the midsoles.**
(TIF)

**S1 Table. Non-diabetes related comorbidities per focus group.**
(PDF)

## Acknowledgments

The authors would like to thank the subjects of the 3 focus groups for their valuable contributions, Leonie Krops for all her efforts as the moderator, and Roosmarijn Geerlings for helping with logistical matters and transcription of some of the focus group discussions.

## Author Contributions

**Conceptualization:** Athra Malki, Gijsbertus J. Verkerke, Rienk Dekker, Juha M. Hijmans.

**Data curation:** Athra Malki, Juha M. Hijmans.

**Formal analysis:** Athra Malki, Gijsbertus J. Verkerke, Rienk Dekker, Juha M. Hijmans.

**Funding acquisition:** Juha M. Hijmans.

**Investigation:** Athra Malki, Gijsbertus J. Verkerke, Rienk Dekker, Juha M. Hijmans.

**Methodology:** Athra Malki, Gijsbertus J. Verkerke, Rienk Dekker, Juha M. Hijmans.

**Supervision:** Gijsbertus J. Verkerke, Rienk Dekker, Juha M. Hijmans.

**Writing – original draft:** Athra Malki.

**Writing – review & editing:** Athra Malki, Gijsbertus J. Verkerke, Rienk Dekker, Juha M. Hijmans.

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
