## [Decision Letter · Decision Letter 0]

6 Sep 2022

PONE-D-22-12625Factors influencing the use of therapeutic footwear in patients with diabetes mellitus and loss of protective sensation. A focus group studyPLOS ONE

Dear Dr. Malki,

Thank you for submitting your manuscript to PLOS ONE. After careful consideration, we feel that it has merit but does not fully meet PLOS ONE’s publication criteria as it currently stands. Therefore, we invite you to submit a revised version of the manuscript that addresses the points raised during the review process.

I would like to sincerely apologise for the delay you have incurred with your submission. We have now received four completed reviews; the comments are available below. The reviewers have raised significant scientific concerns about the study that need to be addressed in a revision.

Please revise the manuscript to address all the reviewer's comments in a point-by-point response in order to ensure it is meeting the journal's publication criteria. Please note that the revised manuscript will need to undergo further review, we thus cannot at this point anticipate the outcome of the evaluation process.

We look forward to receiving your revised manuscript.

Kind regards,

Miquel Vall-llosera Camps

Senior Editor

PLOS ONE

A clean copy of the edited manuscript (uploaded as the new *manuscript* file).

Important: If there are ethical or legal restrictions to sharing your data publicly, please explain these restrictions in detail. Please see our guidelines for more information on what we consider unacceptable restrictions to publicly sharing data: http://journals.plos.org/plosone/s/data-availability#loc

unacceptable-data-access-restrictions. Note that it is not acceptable for the authors to be the sole named individuals responsible for ensuring data access.

Additional Editor Comments:

In general, we would expect qualitative studies to include the following: 1) defined objectives or research questions; 2) description of the sampling strategy, including rationale for the recruitment method, participant inclusion/exclusion criteria and the number of participants recruited; 3) detailed reporting of the data collection procedures; 4) data analysis procedures described in sufficient detail to enable replication; 5) a discussion of potential sources of bias; and 6) a discussion of limitations. Please pay attention to these requirements when revising your manuscript.

Reviewers' comments:

Reviewer's Responses to Questions

**Comments to the Author**

1. Is the manuscript technically sound, and do the data support the conclusions?

Reviewer #1: Yes

Reviewer #2: Yes

Reviewer #3: Yes

Reviewer #4: Yes

2. Has the statistical analysis been performed appropriately and rigorously? 

Reviewer #1: No

Reviewer #2: Yes

Reviewer #3: N/A

Reviewer #4: Yes

3. Have the authors made all data underlying the findings in their manuscript fully available?

Reviewer #1: Yes

Reviewer #2: Yes

Reviewer #3: No

Reviewer #4: Yes

4. Is the manuscript presented in an intelligible fashion and written in standard English?

Reviewer #1: Yes

Reviewer #2: No

Reviewer #3: No

Reviewer #4: Yes

5. Review Comments to the Author

Reviewer #1: Factors influencing the use of therapeutic footwear in patients with diabetes mellitus and loss of protective sensation.

The aim of the present study was to provide insight in the factors that are perceived important in the use of TF for different groups of patients with DM and DPN. To this aim the authors evaluated three different groups, a) patients without therapeutic footwear (TF) and no history of DFU, b) patients without TF and history of DFU and c) patients with TF and DFU.

The study is nicely conducted and the results are nicely presented, despite that, some changes must be made before publication:

INTRODUCTION:

- To long introduction, the authors must shorten it before publication.

STUDY DESIGN AND DATA ALLOCATION

- The authors state in lines 149-151 that it was not possible due to Covid-19 pandemic to assess physical measurements, it could be a bias in the selection of inclusion and exclusion criteria, it must be discussed as a limitation.

- LOPS: authors declare that patients with LOPS were included in the research, but auto perception of LOPS is not a valid method to diagnose DPN and LOPS, authors must explain it in depth and if no neurological assessment was performed, authors might explain how it can be done and if no possible add as a limitation in the discussion section accordingly.

-

RESULTS:

- Higher risk of ulcer occurrence patients seem to have higher knowledge about DFU complications derived from the results presented in table 2, despite this, the qualitative analyses of the results do not let us to reach any conclusion.

- Also, the three studied groups were well balanced in terms of baseline characteristics. That said, the plausibility of the results may be limited by the relatively low number of participants. Was a sample size calculation performed? 8 patients per groups is very small, this need to be included as a limitation of the study.

- The paper has a big weakness due to the lack of quantitative data, for further research, authors should confirm this results in a quantitative study.

DISCUSSION:

- The discussion is coherent, but I recommend shortening it.

Reviewer #2: Thanky you for your article which is very timely in the therapeutic footwear and diabetic foot field.

I have some initial suggestions that may help your manuscript develop further:

-Even though you have made clear that ethics approval was not required, i feel that your work does require ethical approval for an international publication. Therefore can get retrospective approval?

-you have talked about the 'tool' of qualitative research, which I feel is not the right term to use. I also feel it would be nice for you to clarify that you are 'exploring new knowledge generation' which justifies this type of methodology.

- some of your references in the introduction (4, 11,12,13) are quite old. Are there any more recent? if not state that this is a gap in the literature

line 64 - reword. patients who have previously suffered.....for a recurrence of ulceration

line 65 - can you use international guidelines? as well as national?

-after reading the introduction, i am not clear if you are trying to influence design of footwear or influence behaviour change to wear TF. Behavuior change is a different concept and area that you may want to differentiate in this section to make it clear to the reader your intentions.

line 145 - how did you measure TF use in the questionnaire?

Results:

Table 1 is slightly confusing; too many abbreviations, no legend, too many groups of data grouped

table 2 - the first part is a lot of repeated text for the three groups; could you introduce the ranking numbers as another column?

-after reading the main findings, there appears a lot of repetition to each group within the text. This made it difficult to read and to compare and contrast amongst groups. You could condense much of this with a re-working of the presentation.

-your discussion similarly has lots of repetition as you have tried to seperate the groups and report on each. As above suggestion, try to condense and bring your findings and themes together.

your recommendations are very good

Reviewer #3: COMMENT FOR THE AUTHORS

1. Line 30 rather than “do not use therapeutic footwear sufficiently, rewrite as “do not use therapeutic footwear consistently, properly, regularly or as expected.

2. Line 37, “TF & HoU)” should come first to allow for the writing in full for (HoU) history of ulceration

3. Line 37: the content of the bracket should come after the word “discussions”.

4. The presentation of the abstract has not sufficiently provided comprehensive details of the focus of the study which included giving detailed information about the factors that factors that influence or could influence the use of therapeutic footwear. The authors should list the factors based on the outcome of the said focus groups.

5. The use of personal pronoun should be avoided as much as possible in all aspect of the manuscript. Every instance of personal should be rephrased in 3rd person pronoun

Eg: lines 31-32: “If we understand what factors could influence their use” Better rephrased as “If factors that influence their use are understood.”

Instead of “There, we aimed to provide insight in the factors that……” Better rephrased as “Therefore, this study aimed to provide insight in the factors that ……”

6. Line 67: correct “hereby” to “Thereby”

7. Line 81: “Two studies in this review” should be corrected as “Two studies in the review”

8. Line 92: “Currently, we are designing prototype shoes……” should be rephrased as “Currently, the authors of this study are designing prototype shoes …..”

9. Lines 131-134: The groups described should be done properly by differentiated them using numerical number.

10. Detailed information as it relates to the number of participants recruited for the study on the basis of the recruitment channels should be presented and included in the study. That will possibly be of help to future researchers in the aspect of considering recruitment channel for related study design.

11. How many participants were involved in the online questionnaire? This information should be provided.

12. Line 153-154: on how the discussion was organised online? Or “on how the discussion was to be organised” Please rewrite the sentence for clarity.

13. Line: 155: Once all participants joined ….. Should be corrected as “When all participants joined ….”

14. Line: 157; Correct “Hereafter” as “Thereafter”

15. Table 2: The word “patients” should be replaced with “persons” or “subjects”.

16. There is the need for the authors to include more references to support the discussion. The total number of cited references is too limited for an academic paper of this magnitude. The authors should search thoroughly as there are series of published articles in this area of research.

The authors may can consider using the following keywords:

“Therapeutic footwear, Neuropathic Foot, Diabetes”

to search for very current and relevant and updated literatures.

Articles published within the last three years (2019 to date) should form significant aspect in the study to enable the authors provide updated information in the field to the readers.

Reviewer #4: Thank you so much for considering me as a reviewer of this study. This is an article of importance in its field and addressing contemporary issues. Well Done to the Authors!

Some minor comments are proposed in this regard as below.

6. PLOS authors have the option to publish the peer review history of their article (what does this mean?). If published, this will include your full peer review and any attached files.

Reviewer #1: No

Reviewer #2: **Yes: **Richard Collings

Reviewer #3: **Yes: **Stanley I.R. Okoduwa, PhD,

Department of Biochemistry, School of Basic Medical Sciences, Babcock University, Ilishan-Remo, Nigeria.

Reviewer #4: **Yes: **Sayed Ahmed

---

## [Author Response · Author response to Decision Letter 0]

25 Oct 2022

Reviewer #1

6. Introduction

To long introduction, the authors must shorten it before publication.

We agree with the reviewer and, consequently, shortened the introduction.

The initial word count was: 1003, which is currently at: 627.

7. Study design and data allocation

The authors state in lines 149-151 that it was not possible due to Covid-19 pandemic to assess physical measurements, it could be a bias in the selection of inclusion and exclusion criteria, it must be discussed as a limitation.

Agreed, changed in manuscript

8. LOPS: authors declare that patients with LOPS were included in the research, but auto perception of LOPS is not a valid method to diagnose DPN and LOPS, authors must explain it in depth and if no neurological assessment was performed, authors might explain how it can be done and if no possible add as a limitation in the discussion section accordingly.

Agreed, changed in manuscript 

9. Results

Higher risk of ulcer occurrence patients seem to have higher knowledge about DFU complications derived from the results presented in table 2, despite this, the qualitative analyses of the results do not let us to reach any conclusion.

Following the remark, we made changes to table 2 to clarify the information provided in the table.

From (the revised) table 2 one could gather that all groups have sufficient knowledge on diabetes related complications. In fact, the most important complication (DFU) was mentioned by all groups. All groups also knew that therapeutic footwear is prescribed to prevent DFUs. However, the difference between the groups (based on the results of the qualitative study) is that it seems that one has to have experienced an ulcer before evaluating the factor ‘ulcer prevention’ important enough (top 3 in table 2) to start using therapeutic footwear (TF). This means that in table 2 the emphasis for the difference between group-noTF&noHoU, group-TF&noHoU and group-TF&HoU, is on the experience of an ulcer rather than on the risk for an ulcer occurrence. 

The conclusion that is reached from this qualitative study is that it seems that patients have to experience an ulcer before to find ‘ulcer prevention’ important enough to start using TF. Even with experiencing an ulcer before, patients do not rank this factor the highest. This finding emphasizes that besides focusing on the factors that are perceived as important by different groups of persons with DM, one also has to find a way to educate (motivate) persons on the importance of using TF to prevent ulcers.

10. Also, the three studied groups were well balanced in terms of baseline characteristics. That said, the plausibility of the results may be limited by the relatively low number of participants. Was a sample size calculation performed? 8 patients per groups is very small, this need to be included as a limitation of the study.

We agree with the reviewer that for a quantitative study, 8 patients per group would have been too small. However, for a qualitative analysis a group of 8 people is a good size. We based our number of patients per group on literature about qualitative studies. Below you can find studies where it is stated that it is preferred/required to have a group of around 8 for a qualitative study in the form of a focus group.

- https://www.semanticscholar.org/paper/Designing-and-Conducting-Focus-Group-Interviews-Krueger/eb7499b9e559c969d0a8e7bceafd36adc4578eaf

- https://www.fhi360.org/sites/default/files/media/documents/Qualitative%20Research%20Methods%20-%20A%20Data%20Collector's%20Field%20Guide.pdf

- https://pubmed.ncbi.nlm.nih.gov/29262162/

11. The paper has a big weakness due to the lack of quantitative data, for further research, authors should confirm this results in a quantitative study.

We thank the reviewer to bring this point to our attention. We specifically chose a qualitative analysis of the focus group data since we were interested in understanding what factors patients find important (and their importance weight) in using therapeutic footwear and the reasoning (experience, attitude, wishes) behind the importance of the factors for the patients. Qualitative data (gathered from focus groups) on behavior, experience, and attitude can be difficult to quantify, with the possibility of losing the context and narrative behind the qualitative data. According to Tenny et al. one should not quantify data if they are not meant to be quantified. But indeed, we do think that next to a qualitative study (methods such as interviews and focus groups etc.), a quantitative study (based on a quantitative method such surveys and questionnaires etc.) will have an added value. Through a survey one could analyze whether a larger patient population agrees with the ranking of the different factors and if not, to rearrange the rank. One could also ask whether there are some missing factors etc. The study of Tenny et al. (see for the link below) emphasis the importance of both quantitative and qualitative studies and how they can be complementary to each other. https://pubmed.ncbi.nlm.nih.gov/29262162/

12. Discussion

The discussion is coherent, but I recommend shortening it.

We agree with the reviewer and shortened the discussion accordingly.

The initial word count was: 3194, which is currently at: 2470.

Reviewer #2

13. Introduction

Even though you have made clear that ethics approval was not required, i feel that your work does require ethical approval for an international publication. Therefore can get retrospective approval?

Thank you for this suggestion. It may be unclear that with a non-WMO approval we still had to follow the required legal acts and/or guidelines: the Medical Treatment Agreement (WGBO), General Data Protection Regulation (GDPR) and codes of conduct of the FEDERA (Federation of Medical Scientific Institutions). This is why we added the following line to the manuscript: ‘The required legal acts and/or guidelines, the Medical Treatment Agreement (WGBO), General Data Protection Regulation (GDPR) and codes of conduct of the FEDERA (Federation of Medical Scientific Institutions), were followed.’

We do not think a retrospective approval is needed for the following reasons:

The Medical Ethics Committee (METc) of the University Medical Center Groningen is legally authorized to evaluate research that is conducted by the UMCG. The METc decided our study is not subject to the Medical Research Involving Human Subjects Act (WMO). Research is subject to the WMO if the following criteria are met:

1. It concerns medical scientific research and

2. Participants are subject to procedures or are required to follow rules of behaviour.

Our participants were not subject to procedures or were required to follow rules of behaviour. This is why we agree with the decision of the METc that our study does not need a WMO approval. This does not mean that we did not follow other legal acts and/or guidelines. We followed the required legal acts and/or guidelines: the Medical Treatment Agreement (WGBO), General Data Protection Regulation (GDPR) and codes of conduct of the FEDERA (Federation of Medical Scientific Institutions). There has been also several international publications that did not require a WMO approval. Even studies published by PLOS ONE:

- User-relevant factors determining prosthesis choice in persons with major unilateral upper limb defects: A meta-synthesis of qualitative literature and focus group results (europepmc.org) 

- Pre-operative rehabilitation for dysvascular lower-limb amputee patients: A focus group study involving medical professionals | PLOS ONE

14. you have talked about the 'tool' of qualitative research, which I feel is not the right term to use. I also feel it would be nice for you to clarify that you are 'exploring new knowledge generation' which justifies this type of methodology.

Agreed, changed in manuscript

15. some of your references in the introduction (4, 11,12,13) are quite old. Are there any more recent? if not state that this is a gap in the literature

We agree with the reviewer and replaced the references.

16. line 64 - reword. patients who have previously suffered.....for a recurrence of ulceration

We agree with the reviewer, however we removed line 64 to shorten the introduction.

17. line 65 - can you use international guidelines? as well as national?

Thank you for your suggestion. Line 65 was not phrased clearly, the national guidelines follow the ‘International working Group on the Diabetic Foot’ (IWGDF) guidelines. Line 65 is changed in the manuscript and we added the reference of the IWGDF guideline: Guidelines on the prevention of foot ulcers in persons with diabetes (IWGDF 2019 update) (iwgdfguidelines.org)

18. After reading the introduction, i am not clear if you are trying to influence design of footwear or influence behaviour change to wear TF. Behavuior change is a different concept and area that you may want to differentiate in this section to make it clear to the reader your intentions.

We agree and changed the introduction where we now describe more clearly that the goal of our study is to get an overview of factors that are perceived as important for the design of footwear and their importance weight in the use of therapeutic footwear in different patient groups. The end goal (which is outside the scope of this study) is that the results of our study will help in the design of footwear and the selection process of them which in turn may lead to an increase in the use of therapeutic footwear and prevent further deterioration of the patient’s foot condition (e.g. diabetic foot ulcers).

19. line 145 - how did you measure TF use in the questionnaire?

We agree with the reviewer that we should be more explicit. We therefor added a line that the information on the disease and TF use were gathered from the questionnaire through a self-report.

20. Results:

Table 1 is slightly confusing; too many abbreviations, no legend, too many groups of data grouped

Agreed, we changed manuscript accordingly

21. table 2 - the first part is a lot of repeated text for the three groups; could you introduce the ranking numbers as another column?

Agreed, changed in manuscript

22. after reading the main findings, there appears a lot of repetition to each group within the text. This made it difficult to read and to compare and contrast amongst groups. You could condense much of this with a re-working of the presentation.

We agree with the reviewer and shortened the result section by avoiding the repetition the reviewer refers to.

The initial word count was: 5593 (with the patient quotes), which is currently at: 4613 (including the patient quotes).

23. your discussion similarly has lots of repetition as you have tried to seperate the groups and report on each. As above suggestion, try to condense and bring your findings and themes together

We agree with the reviewer and shortened the discussion by avoiding the repetition the reviewer refers to.

The initial word count was: 3194, which is currently at: 2470.

Reviewer #3

24. Line 30 rather than “do not use therapeutic footwear sufficiently, rewrite as “do not use therapeutic footwear consistently, properly, regularly or as expected.

Agreed, changed in manuscript

25. Line 37, “TF & HoU)” should come first to allow for the writing in full for (HoU) history of ulceration

Agreed, changed in manuscript

26. Line 37: the content of the bracket should come after the word “discussions”.

Agreed, changed in manuscript

27. The presentation of the abstract has not sufficiently provided comprehensive details of the focus of the study which included giving detailed information about the factors that factors that influence or could influence the use of therapeutic footwear. The authors should list the factors based on the outcome of the said focus groups.

We agree and changed the abstract where we listed the top 3 ranked factors of each group. We focused on the top 3 ranked factors due to the limited amount of words instructed by the journal.

28. The use of personal pronoun should be avoided as much as possible in all aspect of the manuscript. Every instance of personal should be rephrased in 3rd person pronoun

Eg: lines 31-32: “If we understand what factors could influence their use” Better rephrased as “If factors that influence their use are understood.”

Instead of “There, we aimed to provide insight in the factors that……” Better rephrased as “Therefore, this study aimed to provide insight in the factors that ……”

Agreed, changed in manuscript

The initial sentences of the manuscript (including the background section of the abstract) are changed by Sonja Hintzen, who thoroughly edited our manuscript for language usage, which is why not the exact sentences, that were given as example by the reviewer (lines 31-32), are seen in the background section of the abstract. However, throughout the whole manuscript we rephrased sentences with personal pronoun to 3rd pronoun.

29. Line 67: correct “hereby” to “Thereby”

Agreed, changed in manuscript, however the initial sentence was taken out and rephrased by Sonja Hintzen, who edited our manuscript for language usage.

30. Line 81: “Two studies in this review” should be corrected as “Two studies in the review”

Agreed, changed in manuscript. The initial sentence was taken out and rephrased by Sonja Hintzen, who edited our manuscript for language usage.

However we followed the reviewers comment to change ‘this’ review into ‘the’ review.

31. Line 92: “Currently, we are designing prototype shoes……” should be rephrased as “Currently, the authors of this study are designing prototype shoes …..”

Thank you for the suggestion, however we removed line 92 to shorten the introduction.

32. Lines 131-134: The groups described should be done properly by differentiated them using numerical number.

In the manuscript the three groups were differentiated and shown in Figure 1. The reference to Figure 1 was not done properly (Line 117 gave an error). We made sure the reference to the Figure is corrected.

33. Detailed information as it relates to the number of participants recruited for the study on the basis of the recruitment channels should be presented and included in the study. That will possibly be of help to future researchers in the aspect of considering recruitment channel for related study design.

The researchers of this study were not the main physicians treating the patients. Due to privacy reasons the researcher do not know exactly the number of patients recruited from each channel. 

34. How many participants were involved in the online questionnaire? This information should be provided.

We agree and added this information in the manuscript.

35. Line 153-154: on how the discussion was organised online? Or “on how the discussion was to be organised” Please rewrite the sentence for clarity.

Agreed, changed in manuscript 

36. Line: 155: Once all participants joined ….. Should be corrected as “When all participants joined ….

Agreed, changed in manuscript. The initial sentence was taken out and rephrased by Sonja Hintzen, who edited our manuscript for language usage.

However we followed the reviewers comment to change ‘Once..’ into ‘When..’.

37. Line: 157; Correct “Hereafter” as “Thereafter”

Agreed, changed in manuscript

38. Table 2: The word “patients” should be replaced with “persons” or “subjects”.

Agreed, changed in manuscript. 

In Table 2 we changed the word ‘patients’ to ‘subjects’ since ‘subjects’ is a term that is also used in the PLOS ONE journal guidelines.

39. There is the need for the authors to include more references to support the discussion. The total number of cited references is too limited for an academic paper of this magnitude. The authors should search thoroughly as there are series of published articles in this area of research.

The authors may can consider using the following keywords:

“Therapeutic footwear, Neuropathic Foot, Diabetes”

to search for very current and relevant and updated literatures.

Articles published within the last three years (2019 to date) should form significant aspect in the study to enable the authors provide updated information in the field to the readers.

The reviewer rightfully mentions the number of papers is too limited for the discussion sections. We therefor added a number of relevant references to the discussion section.

Not all the added references are from 2019-2022, however many studies related to the use of therapeutic footwear (and the factors that are related to therapeutic footwear) are older. Even a review from 2019 (see below for the link of the study) where the objective was, to identify in the literature aspects related to the recommendation of health professionals and the use of therapeutic footwear by patients with Diabetes Mellitus, the studies found and were interesting to our study were not very recent.

chrome-extension://dagcmkpagjlhakfdhnbomgmjdpkdklff/enhanced-reader.html?openApp&pdf=https%3A%2F%2Flink.springer.com%2Fcontent%2Fpdf%2F10.1007%2Fs40200-019-00428-9.pdf

Reviewer #4

40. Page 3, Line 56, Please check the space between the reference and the following lines. Also, the spacing is inconsistent in other areas, such as Page 3, line 63 and other references on this page and also throughout the article. Please make them all consistent.

Agreed, changed in manuscript 

41. Page 4, Line 101. Please check grammar for “proofed”.

We agree, however we removed Line 101 in order to shorten the introduction.

42. Page 5, Line 123. Please clarify if the Orthopedic shoe technicians are titled as “Pedorthists” in the Netherlands. If so, please use the appropriate professional title and please check other studies from the Netherlands have been using the term.

Ref: https://onlinelibrary.wiley.com/doi/full/10.1002/dmrr.3237

Agreed, changed in manuscript 

43. Page 6, Line 150, Please clarify if there was an option to collect or verify the patient data such as LOPS, and foot deformities from a verified medical record source or referral source other than the patient’s self-report on them.

We added a number of lines to clarify how the information on the LOPS was gathered. 

44. Page 11, Line 242. Please be consistent with using Capital for “ Appearance” and “prescription process”. 

Agreed, changed in manuscript

---

## [Decision Letter · Decision Letter 1]

12 Dec 2022

PONE-D-22-12625R1Factors influencing the use of therapeutic footwear in persons with diabetes mellitus and loss of protective sensation: a focus group studyPLOS ONE

Dear Dr. Malki,

All four reviewers that have previously reviewed your manuscript agree that the current version is significantly improved, and all previous comments have been satisfactory addressed. The manuscript is technically sound, and it is suitable for publication.

I have additionally asked a fifth reviewer to evaluate your manuscript, and they agree that the comments raised in the previous round of review have been all addressed, and the manuscript is now acceptable for publication. However, a minor comment was raised regarding the referenced exclusion criteria of the study (please see comments bellow). Kindly address the reviewer’s concerns and submit your manuscript for publication.

Thank you very much for responding to each comment; I look forward to receiving your final version.

We look forward to receiving your revised manuscript.

Kind regards,

Dimitrios Sokratis Komaris, Ph.D

Academic Editor

PLOS ONE

Journal Requirements:

Reviewers' comments:

Reviewer's Responses to Questions

**Comments to the Author**

1. If the authors have adequately addressed your comments raised in a previous round of review and you feel that this manuscript is now acceptable for publication, you may indicate that here to bypass the “Comments to the Author” section, enter your conflict of interest statement in the “Confidential to Editor” section, and submit your "Accept" recommendation.

Reviewer #1: All comments have been addressed

Reviewer #2: All comments have been addressed

Reviewer #3: All comments have been addressed

Reviewer #4: All comments have been addressed

Reviewer #5: All comments have been addressed

2. Is the manuscript technically sound, and do the data support the conclusions?

Reviewer #1: Yes

Reviewer #2: Yes

Reviewer #3: Yes

Reviewer #4: Partly

Reviewer #5: Yes

3. Has the statistical analysis been performed appropriately and rigorously? 

Reviewer #1: Yes

Reviewer #2: Yes

Reviewer #3: Yes

Reviewer #4: Yes

Reviewer #5: Yes

4. Have the authors made all data underlying the findings in their manuscript fully available?

Reviewer #1: Yes

Reviewer #2: Yes

Reviewer #3: Yes

Reviewer #4: Yes

Reviewer #5: Yes

5. Is the manuscript presented in an intelligible fashion and written in standard English?

Reviewer #1: Yes

Reviewer #2: Yes

Reviewer #3: Yes

Reviewer #4: Yes

Reviewer #5: Yes

6. Review Comments to the Author

Reviewer #1: Dear authors,

Thank you very much for the responses, all the queries have been addressed consequently.

The quality of the paper improved.

I encourage the authors to continue researching in this field.

Congrats on it.

Reviewer #2: (No Response)

Reviewer #3: (No Response)

Reviewer #4: Well done with the corrections. It is a good article and good area to conduct research on. Keep up the research work to contribute to the scientific community.

Reviewer #5: Reviewers' comments:

Adherence to treatment has been considered as an important factor to influence those clinical outcomes of RCTs. Patients who were adherent had significantly better outcomes than those who were nonadherent. This research plays an important role in discovering new information. The authors have further improved the manuscript according to reviewers’ comments and also explained every question reasonably. However, the following question require further clarification:

1.The exclusion criteria are too simple. Many comorbidities such as cardiac insufficiency, severe hepatic and renal dysfunction, malignant tumor, unsteady gait, lower extremity edema, et.al may influence the choice of the main reason regarding low adherence to use of TF. The characteristics of the subjects should show more information of comorbidities and complications.

7. PLOS authors have the option to publish the peer review history of their article (what does this mean?). If published, this will include your full peer review and any attached files.

Reviewer #1: **Yes: **Dr. Mateo López Moral

Reviewer #2: **Yes: **Richard Collings

Reviewer #3: **Yes: **Stanley I.R. Okoduwa, PhD, [Directorate of Research and Development, Nigerian Institute of Leather and Science Technology, Zaria - Nigeria]

Reviewer #4: **Yes: **Sayed Ahmed

Reviewer #5: No

---

## [Author Response · Author response to Decision Letter 1]

22 Dec 2022

Reviewer 5 comment:

Adherence to treatment has been considered as an important factor to influence those clinical outcomes of RCTs. Patients who were adherent had significantly better outcomes than those who were nonadherent. This research plays an important role in discovering new information. The authors have further improved the manuscript according to reviewers’ comments and also explained every question reasonably. However, the following question require further clarification:

1.The exclusion criteria are too simple. Many comorbidities such as cardiac insufficiency, severe hepatic and renal dysfunction, malignant tumor, unsteady gait, lower extremity edema, et.al may influence the choice of the main reason regarding low adherence to use of TF. The characteristics of the subjects should show more information of comorbidities and complications.

Authors’ response: 

We would like to thank the reviewer to bring this point to our attention. In order to provide more information on the comorbidities and complications, we made changes to Table 1. The most frequent comorbidities in the three focus groups were shown in Table 1. The less frequent comorbidities (extended comorbidities list), are summarized as ‘other’ in Table 1. However, for the interested readers, we also provided another table (S1 Table; table in the ‘supporting information’), with each of the less frequent comorbidity listed per focus group and not summarized as ‘other’.

---

## [Editor Report · Decision Letter 2]

26 Dec 2022

Factors influencing the use of therapeutic footwear in persons with diabetes mellitus and loss of protective sensation: a focus group study

PONE-D-22-12625R2

Dear Dr. Malki,

We’re pleased to inform you that your manuscript has been judged scientifically suitable for publication and will be formally accepted for publication once it meets all outstanding technical requirements.

Kind regards,

Dimitrios Sokratis Komaris, Ph.D

Academic Editor

PLOS ONE
---

## [Editor Report · Acceptance letter]

3 Jan 2023

PONE-D-22-12625R2 

Factors influencing the use of therapeutic footwear in persons with diabetes mellitus and loss of protective sensation: a focus group study 

Dear Dr. Malki:

I'm pleased to inform you that your manuscript has been deemed suitable for publication in PLOS ONE. Congratulations! Your manuscript is now with our production department. 

Kind regards, 

on behalf of

Dr. Dimitrios Sokratis Komaris 

Academic Editor

PLOS ONE